

# 1   Expanding Limits of Laser-Ablation U-Pb Calcite Geochronology

Andrew R.C. Kylander-Clark
Department of Earth Science, University of California, Santa Barbara, CA 93106, USA.
*Correspondence to*: Andrew R. C. Kylander-Clark (akylander@ucsb.edu)
**Abstract.** U-Pb geochronology of calcite by laser-ablation inductively-coupled mass spectrometry (LA-
ICMPS) is an emerging field with potential to solve a vast array of geologic problems. Because of low
levels of U and Pb, measurement by more sensitive instruments, such as those with multiple collectors
(MC), is advantageous. However, whereas measurement of traditional geochronometers (e.g., zircon) by
MC-ICPMS has been limited by detection of the daughter isotope, U-Pb dating of calcite can be limited
by detection of the parent isotope, if measured on a Faraday detector. The *Nu P3D* MC-ICPMS employs a
new detector array to measure all isotopes of interest on Daly detectors. A new method, described herein,
utilizes the low detection limit and high dynamic range of the *Nu P3D* for calcite U-Pb geochronology,
and compares it with traditional methods. A model is created to explore the limits of U, Pb, and U/Pb
ratios that can be measured by LA-ICPMS and can serve as a guide to evaluate potential candidate
materials for geochronology.

## 16   1. Introduction

Calcite U-Pb geochronology by laser-ablation inductively-coupled mass spectrometry (LA-ICPMS) is a
relatively new technique with untapped potential for solving numerous geochronologic problems from the
timing of faulting (e.g., Roberts and Walker, 2016; Nuriel et al., 2017; Goodfellow et al., 2017), age of
ore deposits (Burisch et al., 2017) to paleoclimate, sedimentation, and diagenesis (e.g., Mangenot et al.,
2018; Rasbury et al., 1997; Hoff et al., 1995; Winter and Johnson, 1995; Wang et al., 1998; Rasbury et
al., 1998). Early studies focused on carbonates more likely to contain high concentrations of U, such as
speleothems (e.g., Richards et al., 1998) because the method employed—thermal ionization mass
spectrometry (TIMS)—required weeks to produce reliable ratios; samples with a low likelihood of
success, that is, those with potentially low U contents, were ignored. With the advent of LA-ICPMS,
however, sample throughput and analytical costs have been greatly reduced, such that hundreds of



geoanalytical facilities can, at the very least, screen a large number of samples and choose those suitable
for geochronology in a relatively short period of time and for little cost. Actual analysis time and cost are
also reduced over TIMS—sample preparation is minimal and several samples can be analyzed in a day—
and dozens of labs worldwide have the capability to perform such analyses. LA-ICPMS also has the
advantage of sampling smaller volumes of material; it can thus take advantage of the heterogenous nature
of calcite with respect to U and Pb, using larger datasets to better constrain both the initial $^{207}Pb/^{206}Pb$
compositions and the common Pb-corrected concordia ages. These isochron ages are calculated with ease
on a Tera-Wasserburg diagram similar to other common-Pb-bearing mineral chronometers like titanite
and apatite (e.g., Chew et al., 2014; Spencer et al., 2013).
For typical LA-ICPMS analyses, a 193 nm excimer laser is employed in conjunction with either a single-
collector (SC-ICPMS), or multi-collector (MC-ICPMS) sector-field instrument to take advantage of the
increased sensitivity over a quadrupole (Q-ICP-MS). Traditionally, an MC-ICPMS uses a series of
Faraday detectors on the high-mass side of the detector array to measure $^{238}U$ and $^{232}Th$, and either
Faraday cups or secondary electron multipliers (SEMs) on the low-mass side of the array to concurrently
measure Pb isotopes; SC-ICMPS instruments measure isotope count rates sequentially with a single SEM.
The SC and MC instruments have distinct advantages. Because there is only one SEM on a SC-ICPMS
instruments, there is no need to cross calibrate multiple detectors, yielding simpler data reduction and the
possibility for making 204- or 208-based common-Pb corrections. An MC-ICPMS, on the other hand, is
2–3 times more sensitive than the top SC-ICPMS instruments. This allows precise measurements of
samples with low levels of Pb (i.e., young and/or low common-Pb). Furthermore, its *equivalent* sensitivity
is even higher because it measures all masses at the same time. For example, a SC-ICPMS running only
masses 238, 207, and 206 (232, 208, and 204 are also typically measured) at equal dwell times measures
1/3 the counts over a given cycle than the count rate might suggest because only one mass can be
measured at a time; given that it is also 2-3x less sensitive than an MC-ICPMS, a laser spot must be ~6-9
times bigger to achieve the same precision on a SC-ICPMS. A further advantage of an MC-ICPMS is that





transient signals from changes in U and Pb concentration during ablation affect uncertainties in the
measured $^{207}$Pb/$^{206}$Pb and $^{206}$Pb/$^{238}$U less because all measurements are made concurrently. Finally, the
smaller dynamic range of the SEM can limit samples to a specific range of U concentrations; samples or
reference materials with high U contents can cause the detector to trip to a different measurement mode
(or trip off), yielding spurious results. Low U concentrations in calcite can also be a problem for an MC-
ICPMS measuring $^{238}$U with a Faraday cup because limits of detection are on the order of $10^4$ cps. Even
though the SC-ICPMS has 2–3 times lower sensitivity than an MC-ICPMS, it can precisely measure
count rates of ~$10^2$ cps by employing a secondary electron multiplier (SEM) for all masses. Because of its
reduced sensitivity, however, this equates to a very small range of samples.
Fortunately, a recently introduced MC-ICPMS—the *P3D*—by *Nu Instruments* (Wrexham, UK) can
overcome both of these limitations. The instrument features a Daly detector array that allows for ion
counting on $^{238}$U and the Pb isotopes, and thus expands the range of calcite samples—those with lower U
concentrations—that can be precisely measured by LA-ICPMS. Not only does a Daly detector increase
the sensitivity of the instrument, but unlike a standard SEM, the Daly has a greater dynamic range
(approx. 10-fold over that of an SEM) and can thus be used with a larger range of U concentrations. This
contribution describes the analytical setup for LA-ICPMS using the new *Nu P3D*, comparing the two
modes with each other and with that of a SC-ICPMS, and thereby demonstrating the increased capability
of this new instrumentation to measure calcite U-Pb dates.
**2 Experimental Setup**
The analytical setup is described in Table 1. The instrumentation used in the study consists of a *Photon*
*Machines Excite* 193 nm excimer laser equipped with a HelEx cell, coupled to a *Nu Instruments P3D* for
standard LA-ICPMS analyses. The *Nu Plasma 3D* (*P3D*) contains an array with 6 Daly detectors, 5 on the
low-mass side of the array and 1 on the high-mass side. A 14-Faraday array lies between the Daly
detectors, and allows for measurement of $^{238}$U on either a Faraday or Daly detector, depending on the U
concentration in the sample. Daly detectors are used to measure masses 202, 204, 206, 207, and 208, and





$^{232}$Th is measured on a Faraday cup. Faraday backgrounds yield a 1SD of 0.04 mV, which implies a limit
of detection (LOD) of ~0.1 mV or ~8000 cps; Daly backgrounds yield 1SD of 10–20 cps for isotopes of
Hg and Pb and 1 cps for $^{238}$U, corresponding to LODs of 30–60 and 3 cps, respectively.
In order to compare the difference between SC and MC analytical sensitivities and uncertainties, the laser
was used in conjunction with the *P3D* for two experiments, and an *Agilent 7700* Q-ICPMS for one
experiment. These 3 experiments were run with different spot sizes: *Experiment F*) a 65 µm spot on the
*P3D* using a Faraday for masses 238 and 232, and Daly detectors for masses 208–204 and 202 (110 total
analyses); *Experiment D*) the same configuration, but with 238 measured on a Daly detector; and
*Experiment Q*) a 110 µm spot with the Q-ICPMS and cycle times of 0.06, 0.13, 0.1, and 0.1 s on masses
238, 207, 206, and 204 respectively. During each separate analytical run, each spot was located near the
corresponding spot from the other runs, to minimize uncertainty caused by grain homogeneity. For all
experiments, the laser was run at 10 Hz for 15 seconds and a fluence of approximately 1 J/cm$^2$, yielding a
spot depth of 10–15 µm. Analyses were preceded by two pre-ablation pulses, and 20 seconds of baseline
measurement.

Three calcite samples from the east coast of North America (courtesy of W. Amidon of Middlebury
College) were the main samples measured. These samples—C258, C304, and C273—are ca. 440, 110,
and 80 Ma, respectively, and range in U concentration between a few ppb and a few ppm, with an average
of 120 ppb and a mode of ~20 ppb. Three further samples (C254, C283A, and C283B), were run in
experiments F and D and provide more data for uncertainty comparisons between the two instrumental
configurations (see Figure 1), but the data are described in less detail; they are ca. 440 Ma with variable
Cretaceous (?) (re)crystallization. Calcite and NIST614 reference materials (RMs) were interspersed
every 10 analyses, and a two-stage reduction scheme was employed. *Iolite v.3.0* (Paton et al., 2011) was
used first used to correct the $^{207}$Pb/$^{206}$Pb for mass bias, detector efficiency, instrumental drift etc., and to





correct the $^{238}$U/$^{206}$Pb ratio for instrumental drift, using NIST614 as the primary reference material.
During this first data reduction, 2 seconds were removed from both the beginning and end of both the
RMs and the unknowns, yielding a total count time of 11 s. The $^{238}$U/$^{206}$Pb ratio was then corrected using
a linear correction in *Excel* such that the primary calcite RM, WC-1, yielded 254 Ma (Roberts et al.,
2017) on a Tera-Wasserburg (TW) diagram, anchored to a $^{207}$Pb/$^{206}$Pb value of 0.85. Using this method,
we retrieved ages of 3.01 ± 0.15 (MSWD = 1.3; n = 30) and 65.9 ± 1.1 (MSWD = 1.2; n = 40) for
secondary RMs ASH15 (2.96 Ma; Nuriel et al., in review) and Duff Brown Tank (64 Ma; Hill et al.,
2016), respectively. Analyses with large uncertainties (arbitrarily chosen as 50% for both $^{238}$U/$^{206}$Pb and
$^{207}$Pb/$^{206}$Pb) were discarded; removing these data has little influence on the final age. The data from the
unknowns are all a bit scattered for geological reasons, and were culled to yield single populations for
ease of comparison. (Though beyond the scope of this manuscript, the Paleozoic samples are interpreted
to have suffered partial Pb loss or new crystal growth in the Cretaceous–Tertiary, and the older
Cretaceous sample likely (re)crystallized over an extended period.)
**3 Results**
Table 2 and Figure 2 shows the results for the 6 samples analyzed in the 3 experiments. *Experiment F*
*(P3D – 65 µm spot; U on a Faraday)* yielded ~2.7 mV/ppm of mass 238 on NIST614 and was relatively
stable throughout the run. The sensitivity of *Experiment D (P3D – 65 µm spot; U on a Daly)* was similar
to that of *Experiment F*, but dropped approximately 25% during the analytical session to ~2 mV/ppm of
mass 238 on NIST614. *Experiment Q (Agilent Q-ICPMS – 110 µm spot)* yielded ~110 kcps/ppm of mass
238 on NIST614—equivalent to ~1.8 mV/ppm from a spot ~3 times larger than the 65 µm spot in
experiments 1 and 2—and was stable throughout the run.
For every sample, Experiment F yielded fewer analyses with uncertainties of <50% for $^{206}$Pb/$^{238}$U, as well
as the fewest spots available to make an isochron. These results are consistent with a higher average and
median U ppb; low U concentrations that were measured in Experiments D and Q went undetected or
yielded large uncertainties in Experiment F. Though samples with median $^{238}$U count rates of >10,000 cps





(C273C and C304A) returned fewer viable analyses and worse average $^{238}$U/$^{206}$Pb uncertainties in
Experiment F, the uncertainty of the final age was similar for the higher-U samples on both
configurations on the *P3D*; both yielded lower uncertainties than the Q-ICPMS, despite the 3-fold volume
increase in analyzed material on the Q-ICPMS.
When average count rates of $^{238}$U were below ~8000 cps (near the detection limit of the Faraday detector
on the *P3D)*, however, the number of viable analyses and final age uncertainty was significantly higher in
Experiment D (Table 2 and Figure 2). As an example, sample C258 yielded few viable data points (35%
of the 110 analyses) in Experiment F, fewer than half the number of good analyses in Experiments D and
Q. In addition, the resulting uncertainty in the final age calculation (~4%) is significantly larger than that
of Experiment D, and similar to the resulting uncertainty in Experiment Q (although the Q-ICPMS
yielded >2 times the number of viable spots). Samples C283A and C283C—which also contain low levels
of U—yielded ~50% fewer viable data, necessitated double the average count rates of $^{238}$U, and final
uncertainties that were significantly greater in Experiment F than those of Experiment D.
A summary of the precision vs. U count rate is shown in Figure 1, which shows the precision of $^{238}$U/$^{206}$Pb
and $^{238}$U on a single spot vs. the count rate of $^{238}$U. While there is considerable overlap in the precision vs.
$^{238}$U cps of both $^{238}$U and $^{238}$U/$^{206}$Pb at count rates above approx. 30,000 cps, data collected in Experiment
F yielded no better than a few kcps 2σ uncertainty on $^{238}$U; $^{238}$U/$^{206}$Pb uncertainties consequently show a
similar deviation from the high-count-rate trend. Finally, though the Q-ICPMS shows similar gains in
precision for low-U analyses, the lower sensitivity of the Q-ICPMS results in a minor window of U
concentrations for which analyses have lower uncertainties than those run on the *P3D*.

**4 Discussion**

While there is a clear advantage of using the new Daly-only detector setup on the *P3D* for LA-based
calcite geochronology for some samples, the extent to which this advantage obtains for all samples is still
somewhat ambiguous. The samples that benefit most from the new instrumentation are not only low in U,



but also older. For most measurements of long-lived-isotope geochronology, the analytical limit is
determined by the detection limit of the daughter, not the parent, isotope. However, because older
samples have more daughter product, they are—for samples with low $U/Pb_c$ ratios—analyses of those
samples are more likely to be limited by the count rate of the parent isotope. For samples run on a SC-
ICPMS, this distinction is unimportant because the detection limit of $^{238}U$ is in all cases lower than that
for Pb. However, because the MC-ICPMS has a large sensitivity and precision advantage over the SC-
ICPMS, it is important to distinguish the limits of measurement between the Faraday–Daly and all-Daly
configuration.

**4.1 Theoretical uncertainty of Tera–Wasserburg data**

To explore the limits of precision for each analytical configuration, a dataset was created to represent
different $U/Pb_c$ and $^{238}U$ cps for samples of different ages. Figure 3 shows samples with ages of 440, 80,
and 15 Ma with error ellipses at $U/Pb_c$ ratios of 1, 2, 5, 10, 20, 100 and 200. The size of the ellipse is the
maximum possible uncertainty (from counting statistics only) for a 10s analysis, given the limit of
detection of the instrument. For the all-Daly configuration, the limit of detection is determined by $^{207}Pb$
counts, the least abundant isotope of interest. For this example, 30 cps is assumed (the best achieved
LODs herein; Gerdes et al), but it is important to recognize that the LOD of Pb is based on the
background, which varies from lab to lab, and is also a function of the instrumental sensitivity. For the
Faraday–Daly arrangement, the LOD is limited by $^{238}U$ counts for samples with lower $U/Pb_c$ and by $^{207}Pb$
for samples with high $U/Pb_c$—and increasingly so as the sample age decreases. In this case, a minimum of
30,000 cps of $^{238}U$ is considered—as opposed to the actual ca. 8000 cps LOD—for the Faraday, because
that is the count rate below which a distinct benefit in precision is gained by using the all-Daly
arrangement (see Figure 1 and discussion above). As depicted in Figure 3, older samples yield the greatest
range of $U/Pb_c$ ratios that could yield an advantage of measurement by $^{238}U$ on an ion counter, whereas
the advantage of the Daly detector disappears at $U/Pb_c$ ratios greater than ca. 500 and 250 for samples that
are 80 and 15 Ma, respectively. As an example of the benefit of $^{238}U$ measurement by Daly, an 80 Ma





sample with a maximum U/Pb$_c$ ratio of 10 yields 1400 cps of $^{238}$U at the LOD of 30 cps $^{207}$Pb. Given a
limit of detection of 8000 cps for the Faraday detector, the signal size would need to be 6 times higher
before it could be measured by such means. Furthermore, as discussed above, and shown in Figure 1, the
benefit of the Daly extends to ca. 30,000 cps, or ~20 times the signal that can be measured by the
Faraday–Daly configuration. The benefit extends to 200 times for a U/Pb$_c$ ratio of 1; but some question
arise as to the ability to measure ages at such low U/Pb$_c$ values.
**4.2 Choosing Samples and Instruments**
One intention of this manuscript is to serve as a guide to determine whether any given calcite (or any
other Pb$_c$-bearing) sample is appropriate for U–Pb geochronology, and deciding which type of analytical
equipment to use. As such, the model above is expanded below to explore the U/Pb$_c$ ratios and count rates
needed to produce a reliable age from a given number of analyses. These models are then compared with
the natural results to determine best practices when selecting samples and instruments for analysis.
Calculating theoretical limits is complicated, however, because the uncertainty of an isochron depends on
the distribution of U, Pb$_c$, and thus the distribution of U/Pb and Pb/Pb ratios. For example, a sample with
a given maximum U/Pb$_c$ will yield a final precision that increases with the number of analyses, but this
improvement depends on the distribution of the U/Pb$_c$ ratios. The distribution of U and Pb, and thus
$^{238}$U/$^{206}$Pb and $^{207}$Pb/$^{206}$Pb, in calcite has not been a particular subject of study, but a cursory analysis of
the reference materials and unknowns presented in this manuscript shows that U and Pb concentrations
follow normal distributions; RMs that contain sufficient U (WC-1 and Duff Brown Tank) display a near-
normal distribution of U, whereas the distribution of U concentration of samples and RMs with lower U
contents (ASH15 and unknowns) are log normal (Figure 4). Like U, Hg-corrected $^{204}$Pb counts (a proxy
for common Pb) are normally distributed in RMs and unknowns; $^{208}$Pb counts are similar. The resulting
$^{238}$U/$^{206}$Pb ratios of RMs are normally distributed, but unknowns vary and can be rather uniform (e.g.,
C273C). The manner by which the type of distribution affects the final uncertainty is demonstrated in
Figure 5. The precision of a T-W isochron is best defined by precisely defined end points with maximum





spread; as such, except for samples with extreme U/Pb$_c$, a uniform distribution of $^{238}$U/$^{206}$Pb ratios results
in better final age precision than does a normal distribution. For example, a sample that is 440 Ma with
normally distributed data (and ratios ± 3σ from the mean) requires nearly 2 times as many points to
achieve the same precision as a sample with uniformly distributed data over the same U/Pb range (though
this also depends on the maximum U/Pb$_c$). For normally distributed data with the same maximum U/Pb$_c$,
but only 50% of the spread (i.e., more tightly clustered; Figure 5b), the number of necessary data points
increases further, excepting samples with extreme U/Pb$_c$ (these data would be less dependent on the
precision of the upper intercept).
To compare theoretical data with that obtained from this study—i.e., in order to best represent a natural
dataset—we present and discuss models with 100 uniformly distributed $^{238}$U/$^{206}$Pb data points acquired for
10 s at 10 Hz, recognizing that, as stated above, this is likely a best-case scenario. We explore the
implications of varying maximum U/Pb$_c$ ratios rather than $^{238}$U/$^{206}$Pb ratios because the former are
independent of sample age. The results of the model are shown in Figure 6 Because the precision of
analyses in an ion-counter-only configuration is limited by the count rate of $^{207}$Pb, we calculate the
maximum U/Pb$_c$ ratio that can be achieved for different concentrations of U. For example, a 440 Ma
sample with 10 ppb U run with a 65 µm spot size will yield ~1500 cps of U. The maximum U/Pb$_c$ that
could be achieved with this count rate will be ~13, because any higher values will yield too few counts of
$^{207}$Pb to be measured. Assuming constant U concentration and normally distributed $^{238}$U/$^{206}$Pb ratios, the
best precision on the age of this sample is 0.6%—considerably better than expected for LA-ICPMS (e.g.,
Horstwood et al., 2016). As a comparison, sample C283A contains an average of 10 ppb U (and
maximum of 40 ppb) and thus yields a similar average count rate of $^{238}$U. Its maximum U/Pb$_c$ of 26 is
considerably less than the maximum theoretical value based on the concentration of that particular
analysis because its Pb concentration is well above detection. It should be no surprise then, that the age
uncertainty is higher than the theoretical value at that count rate, but it is also higher than the theoretical
value for a U/Pb$_c$ of 26. Several factors may explain this: 1) though 100 analyses were measured, 32 were





imprecise and rejected; 2) the distribution of $^{238}U/^{206}Pb$ ratios is not uniform; 3) laser instability, detector
response time, laser-induced elemental fractionation (LIEF), signal instability, etc. add uncertainty
beyond that based on counting statistics; and 4) low $U/Pb_c$ values likely have less U and Pb than in the
model.
Although optimistic, this model serves as a guide for the limitation of analyses of calcite by LA-ICPMS,
given U concentration, maximum $U/Pb_c$, and spot size. First, for all but the youngest samples (<<15 Ma),
measurement with the *P3D* can be advantageous for samples with lower U or those necessitating small
spot sizes (e.g., <150 ppb U and <65 µm; <50 ppb U and <125 µm). However, if, for example, the sample
contains concentrations >100 ppb U and the spot can be >100 µm, there is no advantage to using the all-
Daly configuration, and if there is significant material (i.e., spot size can be >200 µm), any LA-ICPMS
will provide the best possible results (that is, the precision will be limited not by the count rate, but rather
other factors such as differences in LIEF, matrix effects etc.). Second, it is highly unlikely that even with
extreme spot sizes and rep rates, that samples with <<1 ppb U can be analyzed. Third, older samples—
when run on the *P3D*—reach their best possible uncertainty with U concentrations of 10–15 ppb; samples
as young as 80 Ma require little more than 30 ppb U, and samples as young as 15 Ma require up to 150
ppb U at moderate spot sizes. Though 2% final uncertainty requires greater concentrations of U for
younger samples (>2500 cps $^{238}U$ are needed for an 80 Ma sample, and >12,000 cps $^{238}U$ for a 15 Ma
sample), it should be noted that—at a given concentration, spot size and $U/Pb_c$—absolute uncertainty is
relatively independent of age; for example, a sample with a 65 µm spot and 10 ppb U yields an
uncertainty of just over 2 Ma, whether the sample is 15, 80, or 440 Ma. Finally, though not depicted
directly in Figure 6, precise ages can be obtained from data with rather low $U/Pb_c$ values. For example,
100 spots with 2% uncertainty yields a final uncertainty of 5–15 Ma (2σ) for samples with $U/Pb_c$ ratios as
low as 1–2. That said, data with such low $U/Pb_c$ ratios should be viewed with caution, as systematic
uncertainties—such as those introduced by inconsistencies in RM isotopic measurements—can lead to
large errors when extrapolating data clustered near the upper intercept.





**4.3 More spots, deeper spots, or bigger spots?**
The theoretical models discussed above use a 10 sec integration time to compare the models to the
empirical data. As discussed above, precision can be improved by increasing the number of analytical
spots, but each spot can also be ablated for longer or at a higher rep rate (i.e., making deeper pits rather
than more pits). One might imagine that these methods might be equally effective, however, there are two
important points to consider. First, individual spot precision is limited to the long-term reproducibility of
down-hole measurements, and is generally no better than 2%; this precision is more difficult to assess in
calcite because most known reference materials exhibit moderate isotopic heterogeneity (e.g., Roberts et
al., 2017). Thus, if increasing the depth of the pit yields analytical uncertainties <2%, then the excess pit
depth is wasted and overall uncertainty fails to improve. Second, whereas increasing the number of spots
leads to a linear increase in the total number of counts (and thus an increase in precision by $\sqrt{n}$), an
increase in pit depth does not lead to a linear increase in counts because ablation yields decrease with pit
depth. Thus, if an increase in total counts could yield better precision, that increase should come from
more, shallower laser pits, rather than fewer, deeper pits.
It is also possible to increase precision by increasing the spot size. In fact, an argument could be made
that a SC-ICPMS that measures 250 µm spots is just as effective as a MC-ICPMS that measures 100 µm
spots. Though this argument has merit, the downside is twofold; 1) some regions of interest are simply
not large enough to permit a spot 2.5X as wide, and 2) U and/or Pb (i.e., U/Pb$_c$) may be heterogeneous at
scales smaller than the spot size, mixing calcite of different age or reducing the range of isotopic ratios
that are used to construct an isochron. Figure 7 demonstrates that even though larger spots can yield a
better per-spot precision, analyzing the same volume of material with smaller spots can yield better age
precision because it can take advantage of the heterogeneous U and Pb concentrations typical of calcite.





## 5 Conclusions


1) Unlike geochronometers with high U and little to no common Pb—such as zircon and monazite—U-Pb
dates of minerals with low U and significant common Pb can be limited by the count rates of the parent
U, rather than the daughter Pb.
2) Given a limit of detection of ~8000 cps for on a Faraday, and the sensitivity of the *Nu P3D*, samples
with as low as 20 ppb U can be analyzed with a 100 μm spot at 10 Hz, and as low as 5 ppb for a 200 μm
spot. Even so, the Faraday is less precise than the Daly at count rates of <30,000 cps, corresponding to U
concentrations of ca. 75 and 20 ppb, with the same respective spot sizes and rep rates.
3) When $^{238}$U is analyzed on a Daly, the limit of detection drops by a factor of >1000, and the analytical
capability is thus limited by the LOD of Pb—$^{207}$Pb in almost all cases—and the ratio required for
optimum precision. The typical LOD of $^{206}$Pb and $^{207}$Pb is ca. 50 cps; it is greater for higher sensitivity
instruments, and those with a higher background of common Pb. For a desired U/Pb$_c$ ratio of ca. 5–10 for
old and young samples, respectively, the required count rate of $^{238}$U would be 500–1000 cps or ca. 5–10
times smaller than can be analyzed on a Faraday detector. The analysis of $^{238}$U on a Daly, therefore
increases the analytical capability to ca. 0.5–2 ppb U for a 100–200 μm spot, respectively.
4) Although the % uncertainty that can be achieved with limited concentrations of U is considerably
different among samples with different ages, the absolute uncertainty is approximately the same. For
example, samples with 1500 cps $^{238}$U yield a maximum possible uncertainty of ca. 2 Ma, nearly
independent of age (older samples yield slightly higher absolute uncertainties). However, because most
LA-ICPMS facilities can achieve up to 2% precision on final age calculations, younger samples can yield
better absolute uncertainties; these can only be achieved at high U concentrations, which limits the
advantage of the *Nu P3D* for young samples.
5) Given enough material and analytical time, a SC-ICPMS, should, in theory, be capable of measuring
samples with concentrations of approximately 2–10 times (i.e., 1–20 ppb U) that of the *Nu P3D*.



However, because of their lower cycle times and inability to make concurrent measurements, SC-ICPMS
instruments likely require considerably higher concentrations of U to obtain comparable date precision.
Figure Captions
Figure 1. Relation between cps $^{238}$U and uncertainty of $^{238}$U (A), and $^{206}$Pb/$^{238}$U (B). The 3 experiments
show the same trend in uncertainty vs. cps at count rates above ~30 kcps 238, but below that, uncertainty
of measurements in Experiment F (238 on the Faraday) increase significantly compared to Experiments D
and Q. Although Experiments D and Q (red and blue symbols) show the similar trends, the sensitivity
gain using the P3D leads to significant improvements in spot uncertainty (large symbols represent
expected uncertainties for a 100 um spot at 10 ppb U).
Figure 2. Tera–Wasserburg concordia diagrams of the 3 unknown samples in each of the 3 experiments.
See text for discussion.
Figure 3. Uncertainty ellipses for each Tera–Wasserburg plot depict the counting uncertainty for 10 s at a
given $^{238}$U count rate for different U/Pb$_c$ ratios of 1, 2, 5, 10, 20, 50, 100, 200. For each U/Pbc, the larger
ellipse is the limit of detection for the all-Daly configuration, or any SC-ICPMS (limited by $^{207}$Pb counts).
The smaller, red ellipse indicates the uncertainty at 30,000 cps $^{238}$U, the point at which the measurement
of $^{238}$U on the Daly is no longer advantageous.
Figure 4. Left-hand plots show the difference in distribution of $^{238}$U/$^{206}$Pb ratios in reference materials and
unknowns; ratios are nomalized to the $^{238}$U/$^{206}$Pb ratio of the age of the sample. Reference materials Duff
Brown and WC-1 have the smallest variation in $^{238}$U/$^{206}$Pb ratios, which correlates well with the
distribution of their U and Pb contents (left-hand plots). Reference material ASH15 and unknown sample
C283C still have a wider log-normal distribution, reflective of their larger distribution of U and Pb
contents relative to Duff Brown and WC-1. Unknown sample C273C has a more uniform distribution of
$^{238}$U/$^{206}$Pb ratios, reflecting its largest distribution of U contents.



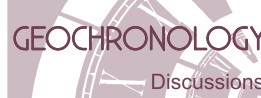 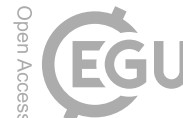

Figure 5. A-C shows an example of the differing randomly generated distributions of 100 analyses with

the same maximum U/Pb$_c$. 5A shows a normal distribution for the entire range of U/Pb$_c$; 5B is a normal

distribution over the upper 50% of the same range. The uniform distribution, shown in 5C, yields the

lowest uncertainties because there are more analyses at both the upper and lower intercepts. D shows how

the percent uncertainty decreases with number of analyses, depending on the type of $^{238}U/^{206}Pb$

distribution depicted in A–C; data in D assumed the best case scenario of 2% uncertainty per data point

and a U/Pb$_c$ ratio of 10 for samples of 440 Ma, 80 Ma, and 15 Ma. Best uncertainties are achieved with

uniform distributions and maximum spread. Although percent uncertainties are always better for older

samples, younger samples yield better absolute uncertainties for well distributed data.

Figure 6. 6A shows the count rate expected with the Nu P3D given for a given spot size at a laser energy

of ~1 J/cm2 and 10 Hz.  Spots indicate analyses of unknowns in Experiment F ($^{238}U$ on the Faraday; color

represents the maximum U/Pb$_c$ ratio—taken from Table 2—and the size represents the uncertainty). Dark

grey area is below the LOD for $^{238}U$ on a Faraday; the light grey area represents $^{238}U$ count rates favorably

measured on the Daly detector. Figures B, D, and F show the maximum possible U/Pb$_c$ ratio that can be

detected, given a minimum count rate of 30 cps for $^{207}Pb$ at different sample ages. C, E, and G show the

best possible uncertainty at the given count rates and spot sizes for 100 analyses, all with the same U

concentration but a uniform distribution of $^{238}U/^{206}Pb$ ratios.

Figure 7. Tera–Wasserburg diagram representing the analysis of a heterogeneous medium using different

spot sizes. Though the bigger spot sizes yield smaller individual uncertainties, the smaller spots take

advantage of the spread in U/Pb$_c$ ratios and thus yield a better overall uncertainty on the lower intercept

age.

Burisch, M., Gerdes, A., Walter, B. F., Neumann, U., Fettel, M., and Markl, G.: Methane and the origin
of five-element veins: Mineralogy, age, fluid inclusion chemistry and ore forming processes in the
Odenwald, SW Germany, Ore Geology Reviews, 81, 42-61, 10.1016/j.oregeorev.2016.10.033, 2017.



Chew, D. M., Petrus, J. A., and Kamber, B. S.: U-Pb LA-ICPMS dating using accessory mineral
standards with variable common Pb, Chemical Geology, 363, 185-199, 10.1016/j.chemgeo.2013.11.006,
346  2014.
Goodfellow, B. W., Viola, G., Bingen, B., Nuriel, P., and Kylander-Clark, A. R.: Palaeocene faulting in
SE Sweden from U-Pb dating of slickenfibre calcite, Terra Nova, 29, 321-328, 2017.
Hill, C. A., Polyak, V. J., Asmerom, Y., and P. Provencio, P. C. T. C.: Constraints on a Late Cretaceous
uplift, denudation, and incision of the Grand Canyon region, southwestern Colorado Plateau, USA, from
U-Pb dating of lacustrine limestone, Tectonics, 35, 896-906, 10.1002/2016tc004166, 2016.
Hoff, J. A., Jameson, J., and Hanson, G. N.: Application of Pb Isotopes to the Absolute Timing of
Regional Exposure Events in Carbonate Rocks - an Example from U-Rich Dolostones from the Wahoo
Formation (Pennsylvanian), Prudhoe Bay, Alaska, Journal of Sedimentary Research Section a-
Sedimentary Petrology and Processes, 65, 225-233, 1995.
Horstwood, M. S. A., Košler, J., Gehrels, G., Jackson, S. E., McLean, N. M., Paton, C., Pearson, N. J.,
Sircombe, K., Sylvester, P., Vermeesch, P., Bowring, J. F., Condon, D. J., and Schoene, B.: Community-
Derived Standards for LA-ICP-MS U-(Th-)Pb Geochronology – Uncertainty Propagation, Age
Interpretation and Data Reporting, Geostandards and Geoanalytical Research, 10.1111/j.1751-
908X.2016.00379.x, 2016.
Mangenot, X., Gasparrini, M., Gerdes, A., Bonifacie, M., and Rouchon, V.: An emerging
thermochronometer for carbonate-bearing rocks: Delta(47)/(U-Pb), Geology, 46, 1067-1070,
10.1130/G45196.1, 2018.
Nuriel, P., Weinberger, R., Kylander-Clark, A. R., Hacker, B., and Craddock, J.: The onset of the Dead
Sea transform based on calcite age-strain analyses, Geology, 45, 587-590, 2017.
Nuriel, P., Wotzlaw, J.-F., Stremtan, C., Vaks, A., and Kylander-Clark, A. R.: The use of ASH15
flowstone as matrix-matched standard for laser-ablation geochronology of calcite, Geochronology, 2, in
review.
Paton, C., Hellstrom, J., Paul, B., Woodhead, J., and Hergt, J.: Iolite: Freeware for the visualisation and
processing of mass spectrometric data, Journal of Analytical Atomic Spectrometry, 26, 2508-2518,
10.1039/c1ja10172b, 2011.
Rasbury, E. T., Hanson, G. N., Meyers, W. J., and Saller, A. H.: Dating of the time of sedimentation
using U-Pb ages for paleosol calcite, Geochimica Et Cosmochimica Acta, 61, 1525-1529, Doi
10.1016/S0016-7037(97)00043-4, 1997.
Rasbury, E. T., Hanson, G. N., Meyers, W. J., Holt, W. E., Goldstein, R. H., and Saller, A. H.: U-Pb dates
of paleosols: Constraints on late Paleozoic cycle durations and boundary ages, Geology, 26, 403-406, Doi
10.1130/0091-7613(1998)026<0403:Updopc>2.3.Co;2, 1998.
Richards, D. A., Bottrell, S. H., Cliff, R. A., Strohle, K., and Rowe, P. J.: U-Pb dating of a speleothem of
Quaternary age, Geochimica Et Cosmochimica Acta, 62, 3683-3688, Doi 10.1016/S0016-7037(98)00256-
380  7, 1998.
Roberts, N. M. W., and Walker, R. J.: U-Pb geochronology of calcite-mineralized faults: Absolute timing
of rift-related fault events on the northeast Atlantic margin, Geology, 44, 531-534, 10.1130/G37868.1,
383  2016.
Roberts, N. M. W., Rasbury, E. T., Parrish, R. R., Smith, C. J., Horstwood, M. S. A., and Condon, D. J.:
A calcite reference material for LA-ICP-MS U-Pb geochronology, Geochemistry Geophysics
Geosystems, 18, 2807-2814, 10.1002/2016GC006784, 2017.
Spencer, K. J., Hacker, B. R., Kylander-Clark, A. R. C., Andersen, T. B., Cottle, J. M., Stearns, M. A.,
Poletti, J. E., and Seward, G. G. E.: Campaign-style titanite U-Pb dating by laser-ablation ICP:
Implications for crustal flow, phase transformations and titanite closure, Chemical Geology, 341, 84-101,
10.1016/j.chemgeo.2012.11.012, 2013.
Wang, Z. S., Rasbury, E. T., Hanson, G. N., and Meyers, W. J.: Using the U-Pb system of calcretes to
date the time of sedimentation of elastic sedimentary rocks, Geochimica Et Cosmochimica Acta, 62,
2823-2835, Doi 10.1016/S0016-7037(98)00201-4, 1998.

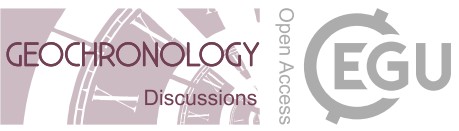

Winter, B. L., and Johnson, C. M.: U-Pb Dating of a Carbonate Subaerial Exposure Event, Earth and
Planetary Science Letters, 131, 177-187, Doi 10.1016/0012-821x(95)00026-9, 1995.





Table 1.
Instrumental parameters of laser-ablation split-stream ICP-MS

|  | MC-ICP-MS | Q-ICP-MS |
|---|---|---|
| Instrument model | Nu Plasma 3D | Agilent 7700x |
| RF forward power | 1300 W | 1300 W |
| RF reflected power | <10 W | <10 W |
| Coolant gas | 13 L/min | 13 L/min |
| Auxiliary gas | 0.8 L/min | 0.8 L/min |
| Make up gas | ~1 L/min | ~1 L/min |
| Monitored masses | $^{238}$U, $^{232}$Th, $^{208}$Pb, $^{207}$Pb, | $^{238}$U(0.06), $^{207}$Pb (0.13), $^{206}$Pb |
| (dwell times listed for *Agilent*) | $^{206}$Pb, $^{204}$Pb/$^{204}$Hg, $^{202}$Hg | (0.1), $^{204}$Pb/$^{204}$Hg (0.1) |
| $^{238}$U sensitivity, dry solution | 0.5% (23 Mcps/ppb) | 0.1% (4 Mcps/ppb) |

|  | Laser-Ablation System |
|---|---|
| Instrument model | Photon Machines Analyte 193 |
| Laser | ATLEX-SI 193nm ArF excimer |
| Fluence | ~1 J/cm² |
| Repetition rate | 10 Hz |
| Excavation rate | ~0.07 um/pulse |
| Spot size | 65–110 μm |
| Delay between analyses | 20 s |
| Ablation duration | 15 s |
| Carrier gas (He) flow (cell; cup) | 0.12; 0.06 L/min |





Table 2. Results from 3 experiments

| sample # | C258 | C273C | C304A | C283A | C283C | C254A |
|---|---|---|---|---|---|---|
| *Experiment F (P3D - 238 on Faraday; 65 μm, ~2.7 mV/ppm U)* | | | | | | |
| total spots | 110 | 100 | 100 | 100 | 100 | 100 |
| $^{238}U/^{206}Pb$ 2σ <50% | 54% | 76% | 63% | 29% | 38% | 63% |
| spots for isochron | 35% | 76% | 47% | 21% | 25% | n/a |
| average U ppb | 40 | 195 | 286 | 28 | 25 | 456 |
| median U ppb | 30 | 73 | 96 | 25 | 27 | 55 |
| average cps 238 | 7100 | 33800 | 46800 | 4600 | 4100 | 73100 |
| median cps 238 | 5300 | 12700 | 15700 | 4100 | 4400 | 8800 |
| avg. $^{238}U/^{206}Pb$ 2σ | 28% | 17% | 17% | 32% | 35% | 24% |
| maximum U/Pb$_c$ | 49 | 145 | 54 | 27 | 17 | n/a |
| Age (Ma) | 437 ± 18 | 80.9 ± 1.5 | 111.1 ± 2.1 | 453 ± 40 | 492 ± 81 | n/a |
| final 2σ | 4.1% | 1.9% | 1.9% | 8.8% | 16.5% | n/a |
| *Experiment D (P3D - 238 on Daly; 65 μm, ~2.1–2.7 mV/ppm U)* | | | | | | |
| total spots | 100 | 100 | 100 | 100 | 100 | 100 |
| $^{238}U/^{206}Pb$ 2σ <50% | 96% | 98% | 97% | 97% | 93% | 90% |
| spots for isochron | 75% | 90% | 84% | 64% | 68% | n/a |
| average U ppb | 24 | 144 | 196 | 11 | 18 | 232 |
| median U ppb | 13 | 59 | 40 | 8 | 18 | 33 |
| average cps 238 | 3800 | 24500 | 27900 | 1600 | 2400 | 29300 |
| median cps 238 | 2100 | 10000 | 5700 | 1200 | 2400 | 4200 |
| avg. $^{238}U/^{206}Pb$ (2σ) | 16% | 10% | 12% | 17% | 16% | 19% |
| maximum U/Pb$_c$ | 30 | 205 | 79 | 26 | 12 | n/a |
| Age (Ma) | 445 ± 11 | 83.5 ± 1.6 | 119.3 ± 2.3 | 430 ± 11 | 430 ± 14 | n/a |
| final 2σ | 2.5% | 1.9% | 1.9% | 2.6% | 3.3% | n/a |
| *Experiment Q (Agilent 7700 Q-ICPMS; 110 μm, ~1.8 mV/ppm U)* | | | | | | |
| total spots | 110 | 100 | 100 | | | |
| $^{238}U/^{206}Pb$ 2σ <50% | 96% | 87% | 94% | | | |
| spots for isochron | 75% | 87% | 94% | | | |
| average U ppb | 28 | 126 | 371 | | | |
| median U ppb | 14 | 68 | 80 | | | |
| average cps 238 | 3100 | 14400 | 39400 | | | |
| median cps 238 | 1600 | 7800 | 8500 | | | |
| avg. $^{238}U/^{206}Pb$ (2σ) | 20% | 14% | 13% | | | |
| maximum U/Pb$_c$ | 26 | 325 | 93 | | | |
| Age (Ma) | 460 ± 18 | 85.4 ± 2.0 | 118.1 ± 4.0 | | | |
| final 2σ | 3.9% | 2.3% | 3.4% | | | |

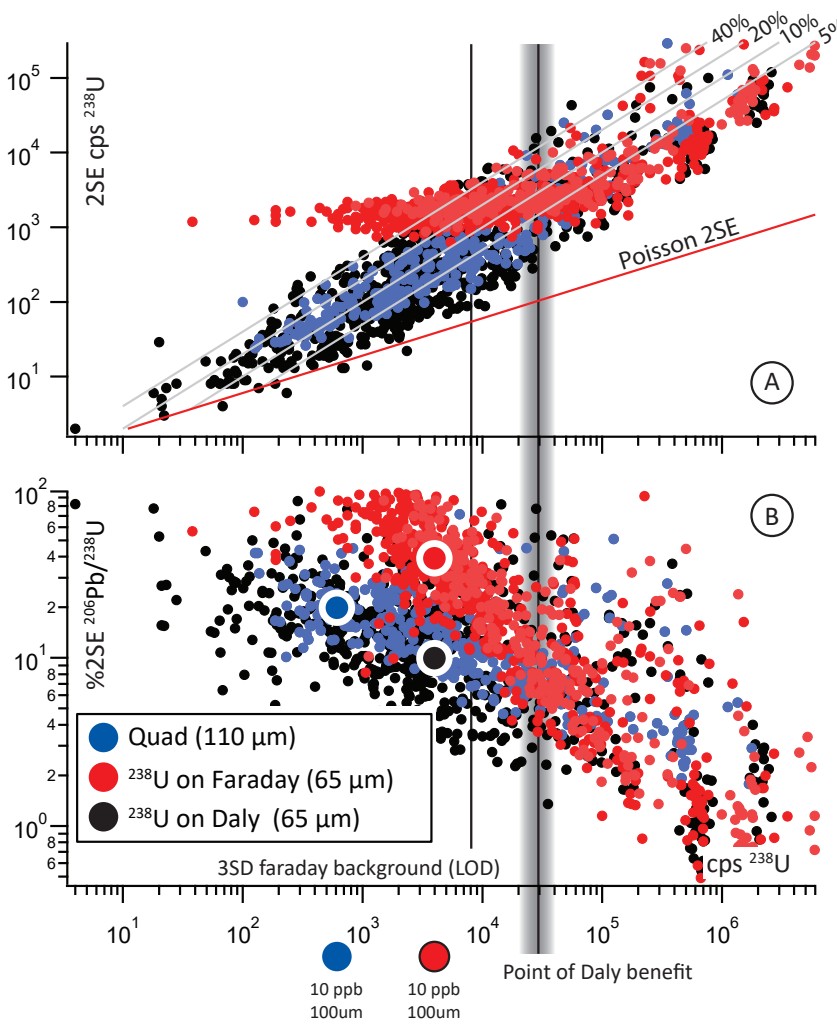

Figure 1. Relation between cps 238U and uncertainty of 238U (A), and 206Pb/ 238U (B). The 3 experiments show the same trend in uncertainty vs. cps at count rates above ~30 kcps 238, but below that, uncertainty of measurements in Experiment F (238 on the Faraday) increase significantly compared to Experiments D and Q. Although Experiments D and Q (red and blue symbols) show the similar trends, the sensitivity gain using the P3D leads to significant improvements in spot uncertainty (large symbols represent expected uncertainties for a 100 um spot at 10 ppb U).



Figure 2. Tera–Wasserburg concordia diagrams of the 3 unknown samples in each of the 3 experiments. See text for discussion.

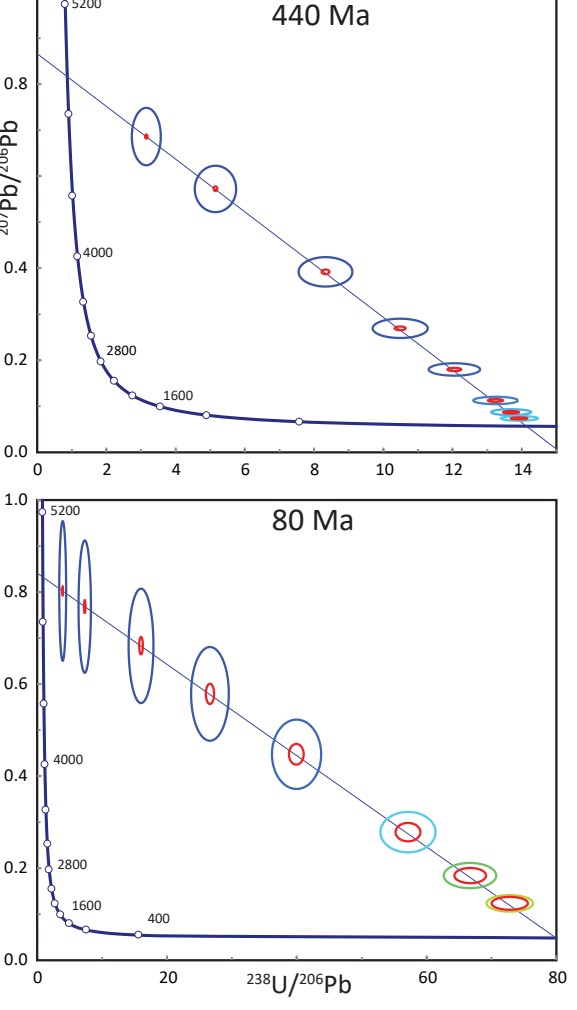

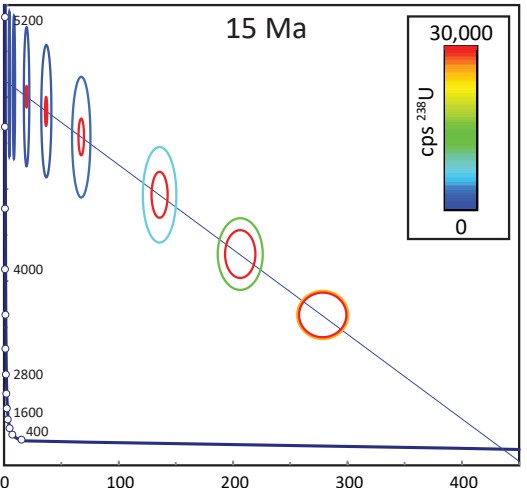

Figure 3. Uncertainty ellipses for each Tera–Wasserburg plot depict the counting uncertainty for 10 s at a given 238U count rate for different U/Pbc ratios of 1, 2, 5, 10, 20, 50, 100, 200. For each U/Pbc, the larger ellipse is the limit of detection for the all-Daly configuration, or any SC-ICPMS (limited by 207Pb counts). The smaller, red ellipse indicates the uncertainty at 30,000 cps 238U, the point at which the measurement of 238U on the Daly is no longer advantageous.




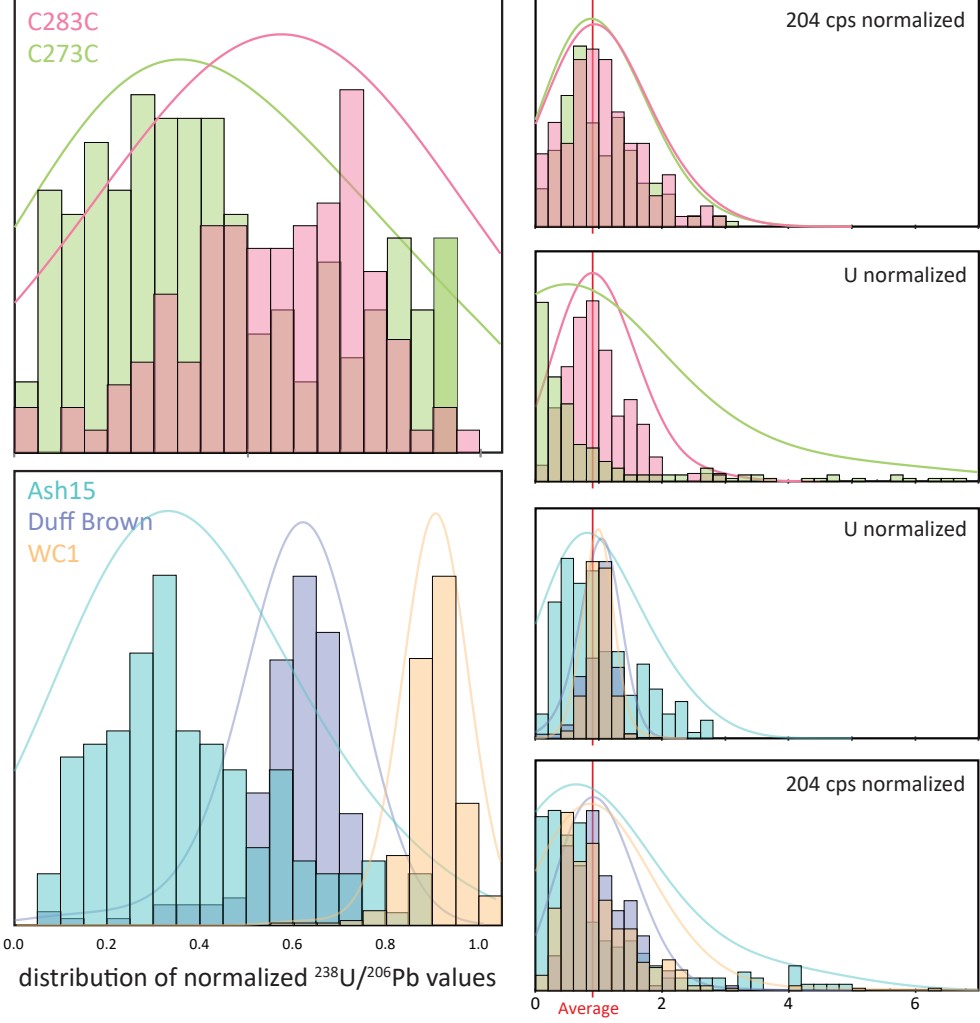

Figure 4. Left-hand plots show the difference in distribution of 238U/206Pb ratios in reference materials and unknowns; ratios are nomalized to the 238U/206Pb ratio of the age of the sample. Reference materials Duff Brown and WC-1 have the smallest variation in 238U/206Pb ratios, which correlates well with the distribution of their U and Pb contents (left-hand plots). Reference material ASH15 and unknown sample C283C still have a wider log-normal distribution, reflective of their larger distribution of U and Pb contents relative to Duff Brown and WC-1. Unknown sample C273C has a more uniform distribution of 238U/206Pb ratios, reflecting its largest distribution of U contents.

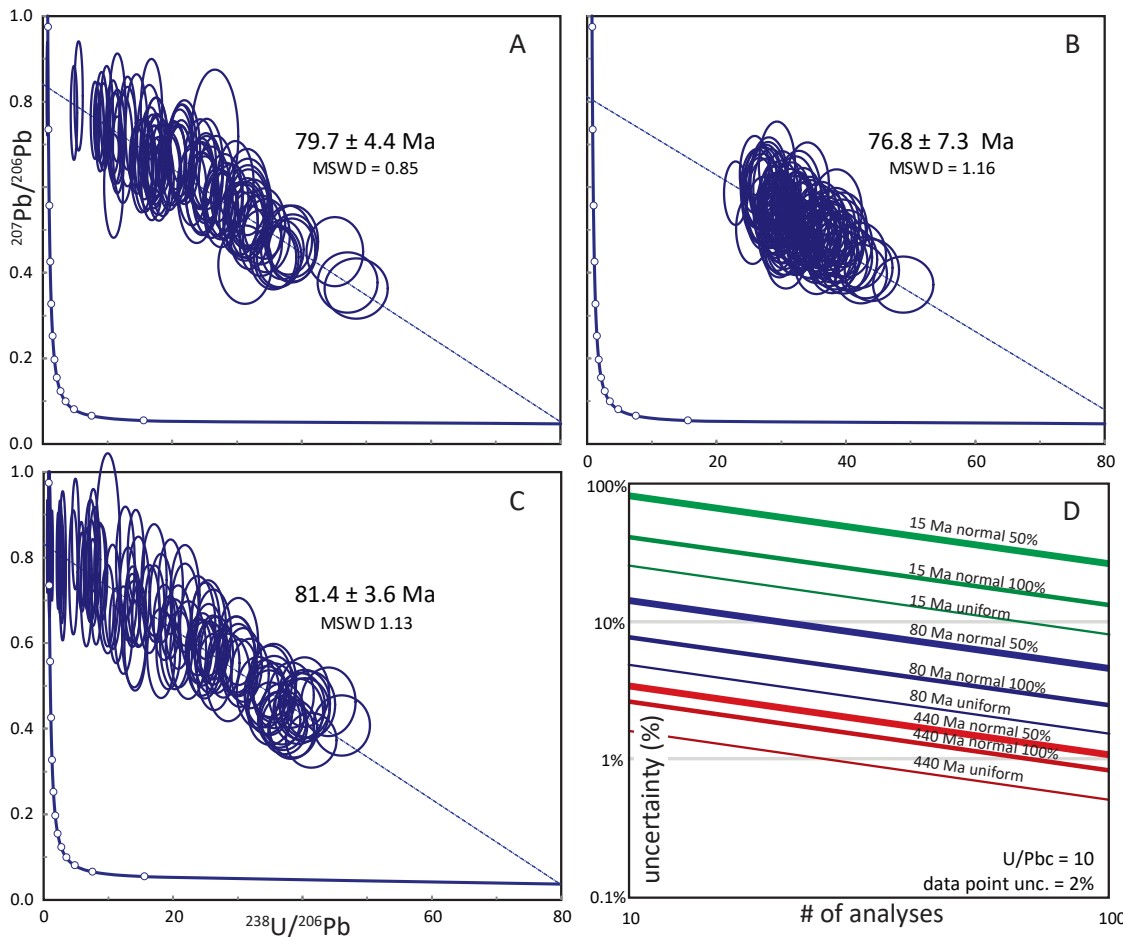

Figure 5. A-C shows an example of the differing randomly generated distributions of 100 analyses with the same maximum U/Pbc. 5A shows a normal distribution for the entire range of U/Pbc; 5B is a normal distribution over the upper 50% of the same range. The uniform distribution, shown in 5C, yields the lowest uncertainties because there are more analyses at both the upper and lower intercepts. D shows how the percent uncertainty decreases with number of analyses, depending on the type of 238U/206Pb distribution depicted in A–C; data in D assumed the best case scenario of 2% uncertainty per data point and a U/Pbc ratio of 10 for samples of 440 Ma, 80 Ma, and 15 Ma. Best uncertainties are achieved with uniform distributions and maximum spread. Although percent uncertainties are always better for older samples, younger samples yield better absolute uncertainties for well distributed data.





Figure 6. 6A shows the count rate expected with the Nu P3D given for a given spot size at a laser energy of ~1 J/cm2 and 10 Hz. Spots indicate analyses of unknowns in Experiment F (238U on the Faraday; color represents the maximum U/Pbc ratio (taken from table 2) and the size represents the uncertainty). Dark grey area is below the LOD for 238U on a Faraday; the light grey area represents 238U count rates favorably measured on the Daly detector. Figures B, D, and F show the maximum possible U/Pbc ratio that can be detected, given a minimum count rate of 30 cps for 207Pb at different sample ages. C, E, and G show the best possible uncertainty at the given count rates and spot sizes for 100 analyses, all with the same U concentration but a uniform distribution of 238U/206Pb

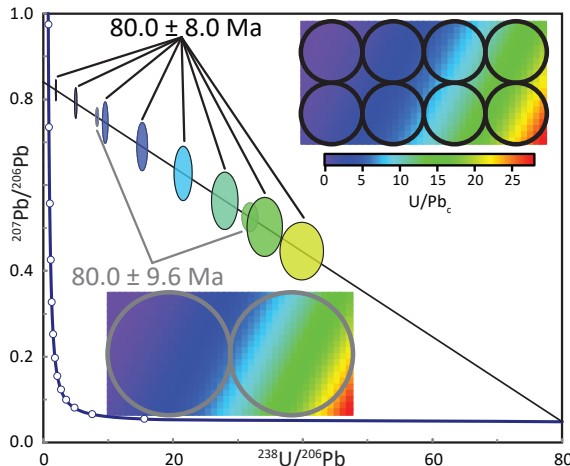

Figure 7. Tera–Wasserburg diagram representing the analysis of a heterogeneous medium using different spot sizes. Though the bigger spot sizes yield smaller individual uncertainties, the smaller spots take advantage of the spread in U/Pbc ratios and thus yield a better overall uncertainty on the lower intercept age.