# Peer review of "Expanding the Limits of Laser-Ablation U-Pb Calcite Geochronology"

_Geochronology, 2020_

## Referee Comment (RC1) · David M. Chew (Referee) · 26 Jul 2020

This paper presents laser ablation U-Pb calcite age data from a new MC-ICP-MS setup with multiple Daly detectors (including on the high mass side for U), and these data are compared with U measured on a Faraday cup on the same setup and by quad-ICP-MS on the same samples. Much of the paper discusses a theoretical model to explore the range of U, Pb, U/Pb ratios and samples ages that can be dated by the new MC-ICP-MS setup (with U on either the Faraday or Daly) and on a quadrupole.

This is a very useful study, such that even if workers in the field do not have access new MC-ICP-MS setup, they will still benefit from the discussion on the U, Pb, U/Pb ratios and samples ages required to give data. It does need some changes though.

The abstract barely mentions the theoretical modelling at all yet discussions based on it are a significant part of the paper, while some of the figures (figures 3 and 6 and to a lesser extent figure 1) require significantly more detailed and clearer figure captions to help the reader. More detail on the modelling would also be welcome, including making the source code available.

The changes to the abstract are important, as the way it is currently written means the reader will not really know what is inside the paper. For example, while many papers have abstracts that are too similar to the conclusions, not one of the conclusions of this study (mainly based on the modelling) features in the abstract. The figure caption changes are definitely required for the reader to clearly follow the discussion on figures 3 and 6 (they will significantly improve its clarity and hence impact).

David Chew

Minor comments

Should the title be "Expanding the limits..." – I think it needs a definite article.

L38-40 "For typical LA-ICPMS analyses, a 193 nm excimer laser is employed in conjunction with either a single collector (SC-ICPMS), or multi-collector (MC-ICPMS) sector-field instrument to take advantage of the increased sensitivity over a quadrupole (Q-ICP-MS)."

I think I will stick up for quads here. Several labs are now doing carbonate dating on the latest generation quads, so this statement is not really true. But it is also confusing, because the quad in this study is then referred to as an SC instrument in lines 80-82. And the quad data presented in this paper are also pretty good – so I would reword L38-40.

L52 you can also have covarying minor changes in U and Pb due to ICP flicker noise – simultaneous MC detection is beneficial for this too.

L60 "Because of its reduced sensitivity, however, this equates to a very small range of
samples." I think I know what you mean here, but is not very clear.

L64 Not everyone will know too much about a Daly detector and maybe a small bit of extra info would be useful here. For example, in addition to its larger dynamic range, I thought they also exhibited more linear counting behaviour over this range?

L92 "Three calcite samples from the east coast of North America" is pretty vague. Veins? Speleothems? A line or two is all that is required.

L103-105 "The 238U/206Pb ratio was then corrected using a linear correction in Excel such that the primary calcite RM, WC-1, yielded 254 Ma (Roberts et al., 2017) on a Tera-Wasserburg (TW) diagram, anchored to a 207Pb/206Pb value of 0.85." A bit more detail needed here on the uncertainty propagation. For example is the initial uncertainty on the Pb ratio of WC-1 ( $\pm$  0.04) propagated through? For the Pb/Pb ratios, is the uncertainty on the NIST glass 207Pb/206Pb propagated through to the unknowns (see Drost et al., 2018, G3).

L115-121 I think it would be simpler to work in cps like in the rest of the paper (but put the mV equivalent in parentheses)

L122-L129. Could be useful to cite a figure here – presumably figure 2 but also figure 1 could be brought back in (it doesn't get much attention when first called out earlier on).

L145-148 "Finally, though the Q-ICPMS shows similar gains in precision for low-U analyses, the lower sensitivity of the Q-ICPMS results in a minor window of U concentrations for which analyses have lower uncertainties than those run on the P3D." I found latter part of this sentence hard to follow; could this minor window also be highlighted on Fig. 1?

Section on "Theoretical uncertainty of Tera–Wasserburg data" L159 Is this a synthetic dataset?

L165 Gerdes et al – what is this?

**GChronD**
L172 – 174 "As depicted in Figure 3, older samples yield the greatest range of U/Pbc ratios that could yield an advantage of measurement by 238U on an ion counter, whereas the advantage of the Daly detector disappears at U/Pbc ratios greater than ca. 500 and 250 for samples that are 80 and 15 Ma, respectively." But figure 3 only shows U/Pbc ratios up to 200....

Section on "Choosing Samples and Instruments" This section is too long not to have sub-headings. I think dsub-headings would really help focus it more.

L1901-191 "The distribution of U and Pb, and thus 238U/206Pb and 207Pb/206Pb, in calcite has not been a particular subject of study". This is not strictly true - Roberts et al. (2020) (Figure 5 therein) do present U and total Pb data from a variety of samples – although admittedly none of their plots exclusively shows intra-sample variability – they are combined inter- and intra-sample variability (i.e. multiple spots from multiple samples). I also think Roberts et al. (2020) should be cited in this study.

L201-L204 – this needs reference to a figure.

L209 "we present and discuss models..." How were these models made? Is the source code available?

L214-L224 again it would help if these samples could be put on one of the figure panels and the figure referred to in this section of text.

L229-L249 – again it would help again if these theoretical discussions could be labelled on the figure panels, with detailed reference to the figures from inside the main text.

Several figure captions are really hard to wrap your head around – more detailed captioning and labelling is required.

Figure 1 – x-axis label unclear – move to below x-axis beside tick marks. What are the lines labelled 40%, 30% etc? Lines of equal RSD%? The U cps values for the large symbols show much higher sensitivities for the MC than the quad – nearly an order of magnitude(?), which appears to contradict the sensitivity differences quoted in lines

GChronD
115-120.

Figure 3. Label panels. I would rephrase along these lines to "Uncertainty ellipses on each Tera–Wasserburg plot depict two end-member type of analyses, with the large ellipses representing the limit of detection for the all-Daly configuration, or any SC-ICPMS (limited by 207Pb counts), and the smaller red ellipses indicates the uncertainty at 30,000 cps 238U, the point at which the measurement of 238U on the Daly is no longer advantageous. The ellipses are coloured according to the 238U count rate, and depict the counting uncertainty for 10 s at this count rate for different U/Pbc ratios of 1, 2, 5, 10, 20, 50, 100, 200."

Figure 6 – panels are not labelled and the figure caption needs to explain what the coloured curves are. This figure caption needs the most work.

**GChronD**

---

## Referee Comment (RC2) · Randy Parrish (Referee) · 19 Aug 2020

Review of Kylander Clark: Expanding Calcite U-Pb dating

This is a paper detailing a methodology to do calcite U-Pb dating in which a very specialised instrument – the Nu Instruments P3D- is used which has multiple Daly ion counting detectors and an array of faraday cups. There are likely to be only a few of these instruments globally and so the main thrust of the paper is to show the specific advantages of this setup and to compare data with other, somewhat less sensitive instruments and detector arrays.

In a sense this paper is about doing 'traditional' calcite dating using a very specialised detector array. Aside from this demonstration of superior sensitivity, there are no par-

ticular advances within the paper that improve the way we do calcite dating, but the performance of this instrument is demonstrated to be reliable and is impressive for its ability to analyse samples with minimal U and Pb. It thus may go some distance to opening up the analysis of very low-U calcite samples at higher spatial resolution to address problems that are otherwise challenging.

Set up of the instrument.

Right away I see a flaw in the set up: there is no ion counter to measure the 232Th signal, only a faraday cup and with noise level of 8000cps, measuring 232Th is thus a write-off with no useful data likely to be collected in a majority of samples. Thus, it is unlikely to be quantified in order to measure radiogenic 208Pb as a contribution to the total 208Pb signal, which is the largest isotope comprising common Pb. I will come back to this.

At the end of this section the author makes a curious statement: Around line 110, The data from the unknowns are all a bit scattered for geological reasons, and were culled to yield single populations for ease of comparison. (Though beyond the scope of this manuscript, the Paleozoic samples are interpreted to have suffered partial Pb loss or new crystal growth in the Cretaceous–Tertiary, and the older Cretaceous sample likely (re)crystallized over an extended period.

When I read statements like this that suggest unknowns are all a bit scattered for geological reasons, it makes me wonder if this is just speculation, with some sort of analytical explanation for the scatter, at least in part, at play, and with the author(s) failing to examine the samples in depth to try to find out the explanation. It is so easy to suggest this sort of thing to explain messy data; in fact there are papers that invoke a wide range of unproven processes for scatter (U-loss, U-gain, Pb loss, variable common Pb composition; recrystallisation, etc.) all of which are just ad hoc explanations for scatter. The best approach, however, is to concentrate on such samples and try to really understand them with more measurements, particularly in their geological, hydrological, and textural context. There is little of this in this paper, largely because it is about methods, but this is an important point that all calcite dating people would do well to take more seriously.

Results

Results are on several reference materials and three different 'unknowns'. The WC-1 calcite is taken as the primary calcite reference material and all results are normalised using a secondary normalisation to the WC-1 calcite; this is standard practise and well documented by earlier papers. The three unknowns are not particularly young – 440 Ma, 120 Ma, and 80 Ma and the reference materials also treated as unknowns are Duff Brown (64 Ma) and Ash (3 Ma). Only WC-1 is very radiogenic; the others have a wide spread in U/Pb ratios. The approach used in all samples is to do regression of an array of spots on the assumption that all measured points are syngenetic and formed at the same time in each sample.

There is a comparison of the success rate and various other parameters that arise from the measurements of samples on the two instruments with the three set-ups. To no surprise, when U is very low, the faraday cup for its measurement performs relatively badly by comparison, but it still does work, to be fair. What is a little bit surprising is how the standard Q-ICP-MS performs so well in comparison to the ideal Daly set up of the Nu P3D, which is illustrated well in the plot of figure 1. What I notice about this figure is that, discounting the experiment with faraday 238U, there is almost complete overlap between the P3D setup with Daly 238U and the Q-ICP-MS, and to some extent a bit more scatter in the P3D data. The Daly is of course better at very high and very low count rates, due to its higher saturation count rate and its lower noise at low count rates, but the advantages of the P3D are not anywhere near as significant as I thought they might be.

The author composes synthetic sample calculations to illustrate the potential strengths and weaknesses of each instrument and setup and then in section 4.2 makes suggestions about which instrument and setup is best suited for unknowns. All of this is interesting, but largely a bit academic. The reality is that people who want to date calcite rarely have the luxury of having an initial session to measure their samples, establish a comprehensive picture of a sample's U and Pb concentrations, radiogenic to common proportions, and then have all of these set-ups available to them to then collect optimized data using the instrument/setup of choice. It will only rarely work that way. More often than not, samples have a geological significance and the challenge is to date as many of these samples as possible, as best as one can given the instrumentation available, and not to have to repeat the work unless necessary to answer ambiguities.

The other reality of all in situ dating, whether this be calcite LA-ICP-MS or accessory mineral SIMS or LA-ICP-MS, is that the Poisson 2SE precision on any ratio is never achievable, and usually with in situ dating, there is a +/-2% barrier that one cannot reduce. Calcite standards also have their own issues with absolute age such as WC-1 with its ∼2% uncertainty in absolute age, and so to some extent the theoretical plots and analysis of this paper are not as applicable in practice as one would like.

The one aspect I cannot find well-described in this contribution is the manner in which the cross calibration of gains and linearity of the multiple Daly detectors have been determined. Has this been done detector by detector using experimental setups that establish ion counting gains and linearity independently? Or, is the use of the WC-1 with its range of count normalisation the method – in other words, if the standard and the other secondary standards give the right data by primary WC-1 normalistion, then all of these linearity and gain issues get accounted for by such a blanket normalisation 'fudge factor'? I suspect it's the latter and frankly it probably works ok, but the authors should be a lot more clear on this, since this is an obvious instrumental issue that is normally detailed in papers that collect multiple ion counter data.

Overall, this is a very careful piece of work with good analysis of the data. It offers some insight into the top level performance of the P3D instruments with its multiple

Daly detectors, and this helps all of us to understand what the benchmarks are as we evolve the techniques. I think the paper should be published largely as is (aside from the comment on ion counters above), because it is very well written and I can find no real issues with what has been done. However, below I comment on what has NOT been done that should be done on a subsequent comparative paper, ie., a followup study.

Additional comments:

Oftentimes in calcite dating, there is little justification in doing it for old samples, and almost all of the action is on younger calcites. This is because there will always be in situ dating uncertainties of +/-2% minimum, and so for a 440 Ma sample this means +/- 10 Ma at least, and many processes that are being studied cannot be resolved when uncertainties get so large. Therefore, the challenge is really to make the method work well for young samples. This study has largely skirted around younger samples but for a methodology to be highly relevant to most geologists, we need the resolution to be a few million years at most; hydrological processes are often fast and subject to disturbance and therefore, revealing tests of methods has to include doing work on very young samples with not so much low U but also very low radiogenic Pb contents. Because textures are complex and often in diagenetic settings, there is no reason to think that all secondary calcite formed at the same time, there is a strong need to be able to calculate single spot ages, like we do in accessory minerals. So far, few studies have done this.

So far the majority of calcite dating studies have ignored 208Pb and 232Th and instead used the T-W diagram for age calculations via regression. This has several problems:

1. Regressions assume all spots are the same age when this may not be the case; scatter is often glossed over

2. 207Pb that is radiogenic is the limiting factor on accuracy and precision. When a sample is young (say 20 Ma) and U is low (0.1 ppm), and when 207Pb background

noise is say 10cps, then the total radiogenic 207Pb might only be a factor of 1 or 2 above this noise, and no matter what setup is used, that will limit precision. Unless these low counts of 207Pb are accurate (and there can be issues when close to background), the ages may not be accurate either. The 207Pb common Pb correction is therefore critical when using the T-W plot, and it has real limitations.

3. This plot fails to use measurements of 232Th or 208Pb, which can be useful. Another way to do this in perhaps a more robust way is to use the 232Th and 208Pb in the plot with Y-axis being the 208Pbcommon/206Pbtotal and the X axis being 238U/206Pbtotal. Calcite rarely has high Th and so the subtraction of 208Pbradiogenic is often trivial, allowing the common Pb correction to be done independently of a radiogenic isotope like 207Pb.

When the author stated early on in the manuscript that data are a bit scattered, I think it is possible if not probable that there may be issues in the measurement of the small 207Pb radiogenic component that could be mitigated if one does not rely upon measurement of 207Pb and instead takes advantage of the 208Pb measurement (virtually all common) and 232Th (usually very low). I discuss this in Parrish et al. (2017) and showed that often this approach works better.

Secondly, readers of this sort of paper should always remember that the field of ICP-MS and particular LA-ICP-MS has seen many orders of magnitude improvement in sensitivity by the use of multiple quads, collision cells, and the like. It is never a good idea to take a standard ICP-MS such as that used in the measurements of this paper, as truly representative of the sensitivity of Q-ICP-MS instruments which can achieve sensitivities nearly as good as MC-ICP-MS in like-for-like experiments. This is just a caution.

I would also love to see the author undertake some testing of individual spot ages using various methods, spot sizes, and so forth to evaluate the power of the P3D instrument to really outcompete Q-ICP-MS in challenging texturally complex samples that require

smaller spot sizes to resolve texturally complex calcite/dolomite growth components. This is where I think the P3D might really have some clear blue water ahead of the other instruments. The application of instrument comparisons on older samples using just the isochron technique is, I think, NOT where the most interesting comparisons of methods of calcite dating are likely to be done. I hope something along these lines might be next project for the author.

R Parrish, 19 August 2020

---

## Author Comment (AC1) · 3 Sep 2020

Thank you for you review; your comments below are much appreciated, and I believe that, by addressing them, the manuscript is significantly improved. I have responded to each comment below and made changes to the manuscript accordingly. Original comments are shown and my responses follow follow each.
* * *
This paper presents laser ablation U-Pb calcite age data from a new MC-ICP-MS setup with multiple Daly detectors (including on the high mass side for U), and these data are compared with U measured on a Faraday cup on the same setup and by quad-ICP-MS on the same samples. Much of the paper discusses a theoretical model to explore the range of U, Pb, U/Pb ratios and samples ages that can be dated by the

new MC-ICPMS setup (with U on either the Faraday or Daly) and on a quadrupole. This is a very useful study, such that even if workers in the field do not have access new MC-ICP-MS setup, they will still benefit from the discussion on the U, Pb, U/Pb ratios and samples ages required to give data. It does need some changes though. The abstract barely mentions the theoretical modelling at all yet discussions based on it are a significant part of the paper, while some of the figures (figures 3 and 6 and to a lesser extent figure 1) require significantly more detailed and clearer figure captions to help the reader. More detail on the modelling would also be welcome, including making the source code available. The changes to the abstract are important, as the way it is currently written means the reader will not really know what is inside the paper. For example, while many papers have abstracts that are too similar to the conclusions, not one of the conclusions of this study (mainly based on the modelling) features in the abstract. The figure caption changes are definitely required for the reader to clearly follow the discussion on figures 3 and 6 (they will significantly improve its clarity and hence impact).

Response: The abstract has been modified to include more of the conclusions of the modelling. Content has been added to the Figures and their respective captions in order to improve their interpretations.

Minor comments Should the title be "Expanding the limits. . ." – I think it needs a definite article.

Response: "the" has been added

L38-40 "For typical LA-ICPMS analyses, a 193 nm excimer laser is employed in conjunction with either a single collector (SC-ICPMS), or multi-collector (MC-ICPMS) sector-field instrument to take advantage of the increased sensitivity over a quadrupole (Q-ICP-MS)." I think I will stick up for quads here. Several labs are now doing carbonate dating on the latest generation quads, so this statement is not really true. But it is also confusing, because the quad in this study is then referred to as an SC instrument

in lines 80-82. And the quad data presented in this paper are also pretty good – so I would reword L38-40."

Response: This is a fair point. We can also employ our quad for many of our calcite analyses. I clarified that SC can be a single collector on either a quad or sector field instrument and deleted the latter part of the sentence.

L52 you can also have covarying minor changes in U and Pb due to ICP flicker noise – simultaneous MC detection is beneficial for this too.

Response: A small line was added to incorporate this important point.

L60 "Because of its reduced sensitivity, however, this equates to a very small range of samples." I think I know what you mean here, but is not very clear.

Response: This section was reworded for clarity.

L64 Not everyone will know too much about a Daly detector and maybe a small bit of extra info would be useful here. For example, in addition to its larger dynamic range, I thought they also exhibited more linear counting behaviour over this range?

Response: This should be true, but there are still similar corrections that need to be applied to both Daly and SEM detectors. Though they are different in design, they function in a similar fashion. The section was reworded to clarify this, and a short bit was added to point out their linearity.

L92 "Three calcite samples from the east coast of North America" is pretty vague. Veins? Speleothems? A line or two is all that is required.

Response: These are fault-related veins (Champlain Valley). This has been noted in the text.

L103-105 "The 238U/206Pb ratio was then corrected using a linear correction in Excel such that the primary calcite RM, WC-1, yielded 254 Ma (Roberts et al., 2017) on a Tera-Wasserburg (TW) diagram, anchored to a 207Pb/206Pb value of 0.85." A bit

more detail needed here on the uncertainty propagation. For example is the initial uncertainty on the Pb ratio of WC-1 ($\pm$ 0.04) propagated through? For the Pb/Pb ratios, is the uncertainty on the NIST glass 207Pb/206Pb propagated through to the unknowns (see Drost et al., 2018, G3).

Response: These are good points and those doing geochronology should be aware of all the uncertainties and how they are treated. In the case of this study, a single value is used for the upper intercepts and NIST values because the idea is to compare the analytical uncertainties rather than use the actual ages of the samples to interpret a specific geologic event. This statement was added in the aforementioned section.

L115-121 I think it would be simpler to work in cps like in the rest of the paper (but put the mV equivalent in parentheses)

Response: Done.

L122-L129. Could be useful to cite a figure here – presumably figure 2 but also figure 1 could be brought back in (it doesn't get much attention when first called out earlier on).

Response: Done. Figure 1 was cited with the first stated observation, and Figure 2 with the last stated observation.

L145-148 "Finally, though the Q-ICPMS shows similar gains in precision for low-U analyses, the lower sensitivity of the Q-ICPMS results in a minor window of U concentrations for which analyses have lower uncertainties than those run on the P3D." I found latter part of this sentence hard to follow; could this minor window also be highlighted on Fig. 1?

Response: I reworded the sentence, referenced Figure 1b and also added a better explanation in the figure caption.

Section on "Theoretical uncertainty of Tera–Wasserburg data" L159 Is this a synthetic dataset?

Response: Yes. I added this to the sentence.

L165 Gerdes et al – what is this?

Response: Proper reference was added.

L172 – 174 "As depicted in Figure 3, older samples yield the greatest range of U/Pbc ratios that could yield an advantage of measurement by 238U on an ion counter, whereas the advantage of the Daly detector disappears at U/Pbc ratios greater than ca. 500 and 250 for samples that are 80 and 15 Ma, respectively." But figure 3 only shows U/Pbc ratios up to 200. . ..

Response: This sentence was reworded to properly represent the figure. Though the figures only show a max U/Pbc of 200, the limitation value was calculated for 80 and 15 Ma. This is also stated.

Section on "Choosing Samples and Instruments" This section is too long not to have sub-headings. I think dsub-headings would really help focus it more.

Response: Sub-headings were added

L1901-191 "The distribution of U and Pb, and thus 238U/206Pb and 207Pb/206Pb, in calcite has not been a particular subject of study". This is not strictly true - Roberts et al. (2020) (Figure 5 therein) do present U and total Pb data from a variety of samples – although admittedly none of their plots exclusively shows intra-sample variability – they are combined inter- and intra-sample variability (i.e. multiple spots from multiple samples). I also think Roberts et al. (2020) should be cited in this study.

Response: Agreed on all accounts – this study is particularly interested in the intra-sample variability, as that pertains to the expected uncertainties on a TW diagram. Roberts et al. has been cited.

L201-L204 – this needs reference to a figure.

Response: Figure 5d is now referenced

L209 "we present and discuss models. . ." How were these models made? Is the source code available?

Response: These models were created using Excel. They are fairly straightforward and can be made available on request. I have added a line to this effect.

L214-L224 again it would help if these samples could be put on one of the figure panels and the figure referred to in this section of text.

Response: Great suggestion. I added a star symbol to Figure 6(a,b,c) and referred to it in the text.

L229-L249 – again it would help again if these theoretical discussions could be labelled on the figure panels, with detailed reference to the figures from inside the main text. Several figure captions are really hard to wrap your head around – more detailed captioning and labelling is required.

Response: Added symbols for the examples given in text, and expanded the legend and caption in the figure. Âň Figure 1 – x-axis label unclear – move to below x-axis beside tick marks. What are the lines labelled 40%, 30% etc? Lines of equal RSD%? The U cps values for the large symbols show much higher sensitivities for the MC than the quad – nearly an order of magnitude(?), which appears to contradict the sensitivity differences quoted in lines 115-120.

Response: Axis label was moved. Lines labeled are percent uncertainties as opposed to absolute (y-axis). This has been clarified in the caption. The sensitivities are shown for the same spot size – the text in 115 describes analyses on the quad at 110 $\mu$m as opposed to 65 $\mu$m on the Nu. The figure shows $\sim$600 cps U for the quad and $\sim$3000 for the Nu (5 fold, rather than an order of magnitude). This is consistent with the text: 600 (cps for Q in Fig 1) *110^2/65^2 (Spot size difference) *2.7/1.8 (mV difference between Q and Nu in text) =$\sim$3000 (cps for Nu in Fig 1)

Figure 3. Label panels. I would rephrase along these lines to "Uncertainty ellipses on

each Tera–Wasserburg plot depict two end-member type of analyses, with the large ellipses representing the limit of detection for the all-Daly configuration, or any SCI-CPMS (limited by 207Pb counts), and the smaller red ellipses indicates the uncertainty at 30,000 cps 238U, the point at which the measurement of 238U on the Daly is no longer advantageous. The ellipses are coloured according to the 238U count rate, and depict the counting uncertainty for 10 s at this count rate for different U/Pbc ratios of 1, 2, 5, 10, 20, 50, 100, 200."

Response: The figure caption was changed and the panels are labeled and examples are given in each panel.

Figure 6 – panels are not labelled and the figure caption needs to explain what the coloured curves are. This figure caption needs the most work.

Response: Panels are now labeled. Examples from the text are inserted and labeled. The caption was expanded to better interpret the figure.

---

## Author Comment (AC2) · 3 Sep 2020

One of the nicest aspects of a journal format such as this one is that the reviewers' comments are visible to everyone. Just as questions that follow conference talks can be as, or more illuminating than the talk itself, reviewer comments can be as insightful as a manuscript. Such is the case here, as the reviewer makes several important points that may otherwise be missed upon revision of the manuscript. I will comment on these points during resubmission, but it is important to note that many of the points will not be properly addressed within the manuscript itself. As such, I encourage others to read these comments (and replies) along with the manuscript. The reviewer comments are listed first, and my responses to each of those comments follow. _______________________________________________________________ Comment: This is a paper detailing a methodology to do calcite U-Pb dating in which a very specialised instrument – the Nu Instruments P3D- is used which has multiple Daly ion counting detectors and an array of faraday cups. There are likely to be only a few of these instruments globally and so the main thrust of the paper is to show the specific advantages of this setup and to compare data with other, somewhat less sensitive instruments and detector arrays. In a sense this paper is about doing 'traditional' calcite dating using a very specialised detector array. Aside from this demonstration of superior sensitivity, there are no par- ticular advances within the paper that improve the way we do calcite dating, but the performance of this instrument is demonstrated to be reliable and is impressive for its ability to analyse samples with minimal U and Pb. It thus may go some distance to opening up the analysis of very low-U calcite samples at higher spatial resolution to address problems that are otherwise challenging. Set up of the instrument. Right away I see a flaw in the set up: there is no ion counter to measure the 232Th signal, only a faraday cup and with noise level of 8000cps, measuring 232Th is thus a write-off with no useful data likely to be collected in a majority of samples. Thus, it is unlikely to be quantified in order to measure radiogenic 208Pb as a contribution to the total 208Pb signal, which is the largest isotope comprising common Pb. I will come back to this.

Reponse: I agree on this point. However, this is a disadvantage to most multi-collector instruments – 232Th is measured on a faraday cup. A further disadvantage is the necessity for proper calibration between ion counters. This is especially difficult when calibrating for 232/208 measurements, because there is no matrix-matched reference material for such a measurement. The 287Pb /207Pb and 208Pb/206Pb can be calibrated against NIST and other glasses, so if Th is negligible (though assessing this with a faraday cup would be difficult as pointed out above), then the 208-based correction can be made. Admittedly, I did not try to make a 208-based correction with this data. However, I have struggled to do so with more success than a TW diagram (i.e., a 207Pb –based correction) for calcite data in the past. I suspect this is due to a combination of: making precise measurements of 232; making an accurate correction for

232/208, and possibly an accurate correction of the 208Pb/206Pb ratio (though NIST glass should provide an accurate reference for this latter correction). I have added an explanation of these points to the analytical setup to make this point.

Comment: At the end of this section the author makes a curious statement: Around line 110, The data from the unknowns are all a bit scattered for geological reasons, and were culled to yield single populations for ease of comparison. (Though beyond the scope of this manuscript, the Paleozoic samples are interpreted to have suffered partial Pb loss or new crystal growth in the Cretaceous–Tertiary, and the older Cretaceous sample likely (re)crystallized over an extended period. When I read statements like this that suggest unknowns are all a bit scattered for geological reasons, it makes me wonder if this is just speculation, with some sort of analytical explanation for the scatter, at least in part, at play, and with the author(s) failing to examine the samples in depth to try to find out the explanation. It is so easy to suggest this sort of thing to explain messy data; in fact there are papers that invoke a wide range of unproven processes for scatter (U-loss, U-gain, Pb loss, variable common Pb composition; recrystallisation, etc.) all of which are just ad hoc explanations for scatter. The best approach, however, is to concentrate on such samples and try to really understand them with more measurements, particularly in their geological, hy- drological, and textural context. There is little of this in this paper, largely because it is about methods, but this is an important point that all calcite dating people would do well to take more seriously.

Reponse: Agreed. As we, as a community, are able to make more precise measurements on smaller aliquots of material, making sense of scatter becomes more of an issue. This is an issue with all geochronometers to some extent; even zircon, one of the most robust chronometers, can yield data that scatter without a reasonable explanation for a particular large-scale geologic process. Rather the scatter can instead be linked to a process on a smaller-scale, such as those mentioned above (U-loss, U-gain, etc.). Calcite is proving to be one of the more complicated geochronometers, and the

community would greatly benefit from greater in-depth study of the particularly compli-cated samples. Such an examination in this case would likely detract from the main point of the paper, nevertheless, I can speculate on these particular calcite veins, as we have dated many from this region. The late Cretaceous sample (the most compli-cated sample) is likely a combination of slightly different vein-filling events. Spots were placed based on pre-analysis that showed regions of high U concentrations, and these groups of spots were scattered across the sample. Many of the samples from this study region yield scatter that correlates to the area of the vein that was analyzed; some of the different regions of the same vein will yield equivalent lower intercepts and different upper intercepts, and others different lower intercepts with similar or distinguishable upper intercepts. This suggests that much of the scatter is caused by multiple calcite-(re)crystallization episodes caused by fluid influx of variable common-Pb compositions. In the case of the Paleozoic sample, there was a similar sample (omitted from this study) that yielded a scatter of data that clearly indicated two (re)crystallization events during the Paleozoic and Cretaceous (clearly distinguishable arrays of data). It is my suspicion that the minor scatter in the Paleozoic samples in this study were caused by minor Pb loss during Cretaceous fluid migration.

Comment: Results Results are on several reference materials and three different 'un-knowns'. The WC-1 calcite is taken as the primary calcite reference material and all results are normalised using a secondary normalisation to the WC-1 calcite; this is standard practise and well documented by earlier papers. The three unknowns are not particularly young – 440 Ma, 120 Ma, and 80 Ma and the reference materials also treated as unknowns are Duff Brown (64 Ma) and Ash (3 Ma). Only WC-1 is very radio-genic; the others have a wide spread in U/Pb ratios. The approach used in all samples is to do regression of an array of spots on the assumption that all measured points are syngenetic and formed at the same time in each sample. There is a comparison of the success rate and various other parameters that arise from the measurements of samples on the two instruments with the three set-ups. To no surprise, when U is very low, the faraday cup for its measurement performs relatively badly by comparison, but

it still does work, to be fair. What is a little bit surprising is how the standard Q-ICP-MS performs so well in comparison to the ideal Daly set up of the Nu P3D, which is illustrated well in the plot of figure 1. What I notice about this figure is that, discounting the experiment with faraday 238U, there is almost complete overlap between the P3D setup with Daly 238U and the Q-ICP-MS, and to some extent a bit more scatter in the P3D data. The Daly is of course better at very high and very low count rates, due to its higher saturation count rate and its lower noise at low count rates, but the advantages of the P3D are not anywhere near as significant as I thought they might be.

Reponse: This is an excellent observation; though I'd like to add a few comments. As can be interpreted from figure 1, when count rates are the same (which requires a considerably larger spot if using the Q-ICPMS), the uncertainty in the Q-ICPMS data is similar to that of the P3D. What is striking about this is that, as mentioned in the manuscript, the actual total counts for each isotope on the Q-ICPMS should be lower than that of the P3D, because each isotope is measured in sequence, whereas the P3D can measure all isotopes simultaneously. I would note that the limit of the precision of the Q-ICPMS data is lower than that of the P3D – this is why the P3D data was placed below the Q-ICPMS data in the figure.

Reponse: An important point should be made that the advantage of simultaneous collection is greatly reduced when the precision is limited by one isotope. For example, in Figure 1B, the 206Pb/ 238U precision is commonly limited by 206Pb, rather than 238U, so one can count longer on 206Pb on the quad, thus reducing the advantage of multi-collection. This has long been known, and I was remiss to not point this out; I added a line in the text to correct this. That said, in the case of this experiment, 232 and 208 were omitted from the cycle to improve the precision (but measured on the P3D). As noted by this reviewer, these isotopes are of interest if one is to make a 208-based correction.

Comment: The author composes synthetic sample calculations to illustrate the potential strengths and weaknesses of each instrument and setup and then in section 4.2

makes sug- gestions about which instrument and setup is best suited for unknowns. All of this is interesting, but largely a bit academic. The reality is that people who want to date calcite rarely have the luxury of having an initial session to measure their samples, establish a comprehensive picture of a sample's U and Pb concentrations, radiogenic to common proportions, and then have all of these set-ups available to them to then collect optimized data using the instrument/setup of choice. It will only rarely work that way. More often than not, samples have a geological significance and the challenge is to date as many of these samples as possible, as best as one can given the instrumentation available, and not to have to repeat the work unless necessary to answer ambiguities.

Reponse: This is a fair point; I would argue that synthetic dataset gives potential users an idea of the differences between the different types of instrumentation, and it can serve as a guide to non-experts. It is duly noted, however, that one cannot expect to achieve such precision as shown in the theoretical calculations, which is why the comparison is made between that and the actual data. It should be noted that the actual data shows the same trends as the theoretical data; this is discussed in section 4.2.2.

Reponse: It is also true that researchers rarely have the luxury of preablating their samples to determine the U,Pb concentrations as a screening tool. This is not always the case, however, and there are certainly cases in which samples can be triaged prior to analysis. The case of these samples is a perfect example: the samples were provided to me as part of a larger project to understand the faulting history of the eastern seaboard. A large number of veins were collected by a professor and his students, and were prescreened as part of several undergraduate projects on a 213 laser coupled to a quadrupole. Though this instrumentation is insufficient to provide accurate dates, it was instrumental in reducing the costly analysis time with the 193 laser/MC-ICPMS. The samples were sent to me with mapped U/Pb values which were rather accurate; analysis near the regions mapped with high U/Pb gave useable data, and analyzing

GChronD

**GChronD**

any other random region almost always gave useless data. This analytical approach yielded far more useful data in a shorter period of time (and much less $$) than analyzing 80-100 random spots by high-precision LA-MC-ICPMS on all the collected samples. Not everyone has this luxury, but it is a technique that can be utilized by many.

Comment: The other reality of all in situ dating, whether this be calcite LA-ICP-MS or accessory mineral SIMS or LA-ICP-MS, is that the Poisson 2SE precision on any ratio is never achievable, and usually with in situ dating, there is a +/-2% barrier that one cannot reduce. Calcite standards also have their own issues with absolute age such as WC-1 with its _2% uncertainty in absolute age, and so to some extent the theoretical plots and analysis of this paper are not as applicable in practice as one would like. The one aspect I cannot find well-described in this contribution is the manner in which the cross calibration of gains and linearity of the multiple Daly detectors have been determined. Has this been done detector by detector using experimental setups that establish ion counting gains and linearity independently? Or, is the use of the WC-1 with its range of count normalisation the method – in other words, if the standard and the other secondary standards give the right data by primary WC-1 normalistion, then all of these linearity and gain issues get accounted for by such a blanket normalisation 'fudge factor'? I suspect it's the latter and frankly it probably works ok, but the authors should be a lot more clear on this, since this is an obvious instrumental issue that is normally detailed in papers that collect multiple ion counter data.

Reponse: I would argue that both the mass bias and detector efficiencies can be corrected with a single standard correction. Both are assumed to be linear corrections, so a single standard measurement can correct for the combination of both factors. That said, the two cannot be determined individually using this method. That is, we cannot determine the mass bias of Pb isotopes or the detector efficiencies, but only the combination of both. Determination of a robust detector efficiency is a rather painstaking process, as peak hopping requires recalibration of zoom lens parameters to ensure equal throughput for any given magnet set point. Hence, I included in the submitted

manuscript a small blurb about the advantage of a SC instrument: "Because there is only one SEM on a SC-ICPMS instruments, there is no need to cross calibrate multiple detectors, yielding simpler data reduction and the possibility for making 204- or 208-based common-Pb corrections," and I further added in section 2, albeit cryptically, "was used first used to correct the 207Pb/206Pb for mass bias, detector efficiency, instrumental drift etc." Nevertheless, I have also added a small bit of text about the lack of Daly detector calibration prior to analyses.

Comment: Overall, this is a very careful piece of work with good analysis of the data. It offers some insight into the top level performance of the P3D instruments with its multiple Daly detectors, and this helps all of us to understand what the benchmarks are as we evolve the techniques. I think the paper should be published largely as is (aside from the comment on ion counters above), because it is very well written and I can find no real issues with what has been done. However, below I comment on what has NOT been done that should be done on a subsequent comparative paper, ie., a followup study.

Additional comments: Oftentimes in calcite dating, there is little justification in doing it for old samples, and almost all of the action is on younger calcites. This is because there will always be in situ dating uncertainties of +/-2% minimum, and so for a 440 Ma sample this means +/- 10 Ma at least, and many processes that are being studied cannot be resolved when uncertainties get so large. Therefore, the challenge is really to make the method work well for young samples. This study has largely skirted around younger samples but for a methodology to be highly relevant to most geologists, we need the resolution to be a few million years at most; hydrological processes are often fast and subject to disturbance and therefore, revealing tests of methods has to include doing work on very young samples with not so much low U but also very low radiogenic Pb contents. Because textures are complex and often in diagenetic settings, there is no reason to think that all secondary calcite formed at the same time, there is a strong need to be able to calculate single spot ages, like we do in accessory minerals. So far,

**GChronD**

Interactive
comment

few studies have done this.

Response: One could argue that this is applicable to all geochronologic studies, and not just those related to calcite, as this minimum commonly applies to most geo-chonometers analysed by LA-ICPMS. Calcite, as pointed out in other studies in this volume, can have larger uncertainties than better behaved minerals, and thus this point is well taken. To refer to an earlier comment, in many cases, researchers do not have the luxury to predetermine the age of their sample, and this technique is the closest they can get to understanding the timing of events. That said, if 2% isn't close enough, they shouldn't use this technique, and will have to look elsewhere to solving their problem.

Reponse: Though the natural samples in this dataset are older than many, they span range from which one could reasonably extrapolate their expectations for younger samples. This is the idea of the modelling work presented; a comparison of natural and synthetic data to give the reader an idea of what to expect for samples of different age and U concentration.

Comment: So far the majority of calcite dating studies have ignored $208Pb$ and $232Th$ and instead used the T-W diagram for age calculations via regression. This has several problems: 1. Regressions assume all spots are the same age when this may not be the case; scatter is often glossed over

Response: Good point: see previous comment on the scatter of the samples presented herein. Note also that one commonly assumes (as the reviewer also pointed out) that the common Pb value is the same for all analyses.

Comment: 2. $207Pb$ that is radiogenic is the limiting factor on accuracy and precision. When a sample is young (say 20 Ma) and U is low (0.1 ppm), and when $207Pb$ background noise is say 10cps, then the total radiogenic $207Pb$ might only be a factor of 1 or 2 above this noise, and no matter what setup is used, that will limit precision. Unless these low counts of $207Pb$ are accurate (and there can be issues when close to

background), the ages may not be accurate either. The 207Pb common Pb correction is therefore critical when using the T-W plot, and it has real limitations.

Reponse: This is an important point to highlight. When 207Pb is near background, the background corrected value may not just be imprecise, but because of possible interferences, the value could be inaccurate.

Comment: 3. This plot fails to use measurements of 232Th or 208Pb, which can be useful. Another way to do this in perhaps a more robust way is to use the 232Th and 208Pb in the plot with Y-axis being the 208Pbcommon/206Pbtotal and the X axis being 238U/206Pbtotal. Calcite rarely has high Th and so the subtraction of 208Pbradiogenic is often trivial, allowing the common Pb correction to be done independently of a radiogenic isotope like 207Pb.

Reponse: These are all excellent points. The downside, not mentioned here, is twofold: 1) measurement of the 232/208 ratio is required for samples that contain Th (that is, one needs to correct for the mass bias, etc., of Th/Pb without a matrix-matched standard), and 2) one needs to calibrate the detectors for 208Pb/206Pb. This latter point could be done using NIST, as is done for the 207Pb/206Pb.

Comment: When the author stated early on in the manuscript that data are a bit scattered, I think it is possible if not probable that there may be issues in the measurement of the small 207Pb radiogenic component that could be mitigated if one does not rely upon measurement of 207Pb and instead takes advantage of the 208Pb measurement (virtually all common) and 232Th (usually very low). I discuss this in Parrish et al. (2017) and showed that often this approach works better.

Reponse: When mentioning the possibility of performing a 208-based in the introduction, I failed to reference this paper. This has been rectified in the revised version.

Comment: Secondly, readers of this sort of paper should always remember that the field of ICPMS and particular LA-ICP-MS has seen many orders of magnitude improvement in sensitivity by the use of multiple quads, collision cells, and the like. It is never a good idea to take a standard ICP-MS such as that used in the measurements of this paper, as truly representative of the sensitivity of Q-ICP-MS instruments which can achieve sensitivities nearly as good as MC-ICP-MS in like-for-like experiments. This is just a caution. I would also love to see the author undertake some testing of individual spot ages using various methods, spot sizes, and so forth to evaluate the power of the P3D instrument to really outcompete Q-ICP-MS in challenging texturally complex samples that require smaller spot sizes to resolve texturally complex calcite/dolomite growth components. This is where I think the P3D might really have some clear blue water ahead of the other instruments. The application of instrument comparisons on older samples using just the isochron technique is, I think, NOT where the most interesting comparisons of methods of calcite dating are likely to be done. I hope something along these lines might be next project for the author.

Response: Another fair point – natural samples rarely reveal themselves as ideal candidates to serve as examples for the geologic community. This suggestion is certainly one of our aims going forward, as we continue to strive to create the ideal instrument configurations for geologic endeavors.

---

## Author Response (AR1)

**UNIVERSITY OF CALIFORNIA, SANTA BARBARA**

[Figure]

DEPARTMENT OF EARTH SCIENCE

SANTA BARBARA, CALIFORNIA 93106-9630

September 23, 2020

Dr. Klaus Mezger
Editor, Geochronology
EGU Publications

Dear Dr. Mezger,

Thank you for editing the manuscript "Expanding the Limits of Laser-Ablation U-Pb Calcite Geochronology." I much appreciated the fruitful discussion with the reviewers and am hopeful that this paper will provide many readers with a reference with which to review when considering geochronometers with low radiogenic/common Pb ratios. I am submitting a final version with only one last correction – Nuriel et al., in review, has been changed to Nuriel et al., 2020. The only other suggested correction was to move the reported ages of the samples to the results section. Though I agree that would be standard practice for a paper in which the ages had geologic significance, but in this case, the significance of the ages is unimportant, but rather the U, Pb, concentrations and uncertainties (and how those relate to the ages). I thus believe it is easier to follow if the ages are reported in the sample descriptions, rather than in the results; after all, they were chosen as examples for this work because I knew beforehand that they had different ages. If you have any further questions or concerns, please do not hesitate to contact me.

Sincerely,

Andrew Kylander-Clark

[revised manuscript text omitted]